# Physico–Chemical Interaction between Clay Minerals and Albumin Protein according to the Type of Clay

**Hyoung-Mi Kim [1] and Jae-Min Oh [2],\***

[1]  Department of Chemistry, Yonsei University, Seoul 26493, Korea
[2]  Department of Energy and Materials Engineering, Dongguk University-Seoul, Seoul 04620, Korea
**\***  Correspondence: jmoh.nbml@gmail.com; Tel.: +82-02-2260-4977

**Abstract:** Clay minerals are widely utilized in pharmaceutical and dermatological sciences as a gastrointestinal medicine or skin remediation agent. In order to verify the feasibility of clays as an injection, pill, or topical agent, it is important to study their interactions with biological components, such as proteins. In this study, we utilized a protein fluorescence quenching assay and circular dichroism spectroscopy to evaluate general aspects of protein denaturation and conformational change, respectively. Three different clays; layered double oxide (LDO), montmorilonite (MMT) and halloysite nanotube (HNT), were treated with albumin and the physico-chemical effect on the protein's conformation was investigated. MMT was shown to influence the conformational change the most, owing to the large accessible adsorption site. HNT showed meaningful circular dichroism (CD) band collapse as well as fluorescence quenching in the protein, suggesting a potential harmful effect of HNT toward the protein. Among the three tested clays, LDO was determined to affect protein structure the least in terms of three-dimensional conformation and helical structure.

**Keywords:** clay minerals; albumin; montmorillonite; halloysite nanotube; layered double oxide; circular dichroism; protein fluorescence quenching

## 1. Introduction

Clay minerals and engineered clay materials have long been utilized in a variety of applications. Natural clays with water absorption and proton scavenging properties can be administered as gastrointestinal medicines such as laxatives or antidiarrhea agents [1–3]. Due to their high absorption/adsorption properties, cationic exchange capacity and antibacterial effect, several smectite clays have been dermatologically utilized as mud pack materials, spa ingredients and skin remediation agents [2,4,5]. Industrially, clays are utilized in order to improve either chemical or mechanical properties of already-existing materials. Catalytic molecules can be stabilized on the large surface of the 2-dimensional layer of clays to maximize the surface exposure of their active sites and to optimize their catalytic activity [6–8]. Several clays can be delaminated in a polymer matrix and thus have been reported to enhance the mechanical strength of polymer materials [9,10] and to control gas permeation properties through the tortuous pathway [11–13]. In this regard, polymer-clay nanocomposites are utilized as shock absorbing bumpers in automobiles or as active packaging materials for specific foods. Furthermore, recent advances in nanotechnology and biotechnology have opened a new era of clays as reservoirs or carriers for bio-active molecules. Smectite clays with cationic exchange capacity have been reported to accommodate amine-containing drugs for safe storage and controlled release [14]. Layered double hydroxides (LDHs) with anionic exchange capacity are known to stabilize drugs with carboxylate or phosphate moieties for efficient cellular delivery [15–17]. Furthermore, clays have been known to accommodate biopolymers, like nucleic acids, through electrostatic forces [18–22], ligand exchange [18,19] and hydrogen bonding [18,19].

Recent pioneering approaches suggested that proteins—large and complicated biomolecules—can be stabilized by clays [19,23–25], expanding the applicability of clays as bio-sensors or protein therapy platforms. It is possible for clays to accommodate proteins in their intra- or inter-particular space. Tubular clays, like halloysite nanotubes (HNT), can accommodate proteins inside the tube lumen [26–29]. When proteins are incorporated into clays, there are concerns that the conformation of the protein may change, as the three-dimensional shape is strongly related to the function of the protein. The primary structure of proteins, the sequence of amino acids, is maintained by covalent bonds. The secondary structure, such as the alpha helix or beta sheet, is relatively stabilized via periodic hydrogen bonds. Several pioneering works showed that the protein's secondary structure can be affected by a variety of conditions, such as the solution type, the clays' morphology, etc., based on Fourier transform infrared (FT-IR) and circular dichroism (CD) spectroscopies [26,27]. Compared with the primary and secondary structure, the tertiary structure, which determines the function of the protein by controlling the three-dimensional structure, is formed by relatively weak forces, such as van der Waals interactions, occasional disulfide bonds, etc., and thus the tertiary structure can be influenced by external chemical and physical stimuli. There have been reports that complicated denaturation phenomena of proteins upon reaction with nanoparticles can be quantitatively evaluated by the protein fluorescence quenching assay [28–30], although we cannot clarify the origin of denaturation with this method. In order to find biomedical applications of clays, one must evaluate how the chemical and physical properties of clays affect both the secondary structure and general 3-dimensional structure of proteins.

In order to examine the effect of the physicochemical properties of clay on protein structural change, albumin, one of the ubiquitous and most abundant proteins in mammals, was chosen as a model protein. Three different clays—montmorillonite (MMT), layered double oxide (LDO) and HNT—were selected, considering the physico-chemical properties of clays. MMT has a chemical composition of $(Na,Ca)_{0.33}(Al,Mg)_2(Si_4O_{10})(OH)_2·nH_2O$, and is structured as a three-layer package; one layer of Al octahedron sandwiched between two layers of $SiO_4$ tetrahedra. The particles are plate-shaped and are composed of layers with an average diameter around 1 μm and thickness of 0.96 nm. MMT is known to have a negative surface charge with a cationic exchange capacity of 92.6 milliequivalent/100 g. LDO is obtained by calcination of LDH, where the general formula of LDH is $[M(II)_{1-x}M(III)_x(OH)_2]^{x+}[A_n^{-x/n}·yH_2O]^{x-}$ (M(II): divalent metal, M(III): trivalent metal, $A^{n-}$: $n$-valent anion) [17]. Upon heat treatment, dehydration, dehydroxylation and evaporation of LDH occur, resulting in $(M(II)O)_{1-x}(M(III)O_{1.5})_x$. LDO can recover the LDH structure upon addition of water and appropriate anionic species [14]. HNT is an aluminosilicate clay mineral with the chemical formula $Al_2Si_2O_5(OH)_4$, It has a small cylindrical morphology with an outer diameter of 50–60 nm, an inner diameter of 12–15 nm, and a length of 0.5–10 μm. Due to its morphological features, it is often referred to as nanotube.

In this study, we utilized several methods to investigate change in the three-dimensional conformation of proteins upon treatment with clays. First, scanning electron microscopy was utilized to visualize the surface morphology of the clay–protein interactant. Then, X-ray diffraction was applied to check if there was structural change in clays that might be related to the molecular level interaction between clays and proteins. Then, spectroscopic methods [30–33], including the protein fluorescence quenching assay and circular dichroism spectroscopy, were applied to evaluate general aspects of protein denaturation and secondary structural change, respectively. The conformational change of the protein will be discussed in terms of physico–chemical properties of clays and reaction temperatures.

## 2. Materials and Methods

### 2.1. Materials

Bovine serum albumin (BSA), HNT and MMT clay were purchased from Sigma-Aldrich, USA. The LDH, pristine of LDO, was synthesized by the co-precipitating method using metal solution (0.015 mol of aluminum nitrate and the 0.0284 mol of magnesium nitrate in deionized water) and alkaline

solution (0.9 mol/L of sodium hydroxide and 0.675 mol/L of sodium bicarbonate). The alkaline solution was added to the metal solution using the dropping funnel until the pH was 9.5. After titration, the reactant was aged for 24 h and then hydrothermally treated for 24 h. The powdery LDH was calcined at 400 °C for 8 h to obtain LDO.

### 2.2. Preparation of the Protein–Clay Interactant

In order to confirm interaction between BSA and two types of clays, HNT and MMT, 1 mg/mL of clay suspension was prepared in deionized water and then the equivolume of 1 mg/mL of BSA solution was added. Protein interaction with LDO was carried out by reacting BSA solution (1 mg/mL) and LDO powder (1 mg) directly.

### 2.3. Surface Morphology and Crystal Structure Analyses

X-ray diffraction (XRD) patterns of clays before and after the protein interaction were recorded on an X-ray diffractometer (Bruker D2 Phaser, Berlin, Germany) using Cu Kα radiation ($\lambda = 1.5106$ Å). Data were collected from the 2θ range of 5°–70° with a scan rate of 2°/s and at 30 kV and 10 mA. Scanning electron microscopy (SEM, FEI Quanta FEG250, Hillsboro, OR, USA) was used to characterize the size and morphology of samples.

### 2.4. Quantification of Adsorbed Protein on Clays

The adsorbed amount of BSA on each clay was quantified by measuring the initial and final concentration of BSA in the reaction supernatant utilizing the Bradford assay. As described in Section 2.2, 1 mg/mL of clay suspension was mixed with 1 mg/mL of BSA solution, which was incubated for 24 h at 37 °C. The supernatant containing BSA was collected by filtering (syringe filter; cellulose membrane). Then 5 mL of supernatant was mixed with 0.1 mL of Bradford dye solution (Biorad). The mixture was incubated at room temperature for 5 min and absorbance at 595 nm was measured using a microplate reader (Varioskan LUX, ThermoFisher Scientific, Waltham, MA, USA).

### 2.5. Protein Fluorescence Quenching Assay

The different concentrations of clay suspensions (2, 1, 0.8, 0.6, 0.5, 0.4, 0.2, 0.1, 0.05 mg/mL) were prepared. Then 1 mL of BSA solution (2 mg/mL) was mixed with 1 mL of the clay suspension in Dulbecco's Phosphate-Buffered Saline (DPBS; pH 7.4). The mixture was stirred for 3 h and then 200 μL of sample was collected. The suspension was subjected to fluorescence measurements (excitation wavelength 280 nm, emission wavelength 340 nm) in a microplate reader (Varioskan LUX, ThermoFisher Scientific).

In order to check time and temperature dependent protein denaturation upon interaction with clays, we monitored the fluorescence quenching ratio along a time course at the temperatures 25 °C and 60 °C. Specifically, an equivalent volume of BSA solution (1 mg/mL) and clay suspension (1 mg/mL) was mixed and the quenching ratio from 20 min to 24 h was measured.

The fluorescence quenching ratio in this study stands for the $(1 - I/I_0) \times 100\%$, where $I_0$ and $I$ refer to fluorescence emission intensity without and with quencher, respectively.

### 2.6. Circular Dichroism Spectroscopy

Circular dichroism (CD) spectra were recorded on a Chirascan plus (AppliedPhotophysics) by the use of a high-performance UV-Vis-IR avalanche photo-diode and fluorescence monochro detector. Each spectrum represents the average of three scans obtained by collecting data at a scan speed of 100 nm/min. The sample was prepared by mixture of 1 mL of BSA solution (1 mg/mL) and 1 mL of clay suspension (1 mg/mL) in DPBS (pH 7.4). The interaction temperature was set at 25 °C and 60 °C, respectively, to evaluate the effect of temperature on denaturation.

## 3. Results

### 3.1. Protein–Clay Interactant: Surface Morphology and Crystal Structure

As shown in Figure 1, protein molecules are expected to be adsorbed on the surface or in the inter-particle space of clays. According to previous reports, interactions between clays and organic moieties can be electrostatic attractions [24,32,34–37], hydrophobic–hydrophobic interactions [25,34], hydrogen bonding [34,36], ion exchange [25] and van der Waals forces [34,36], with all these interactions occurring at the surface of clay layers.

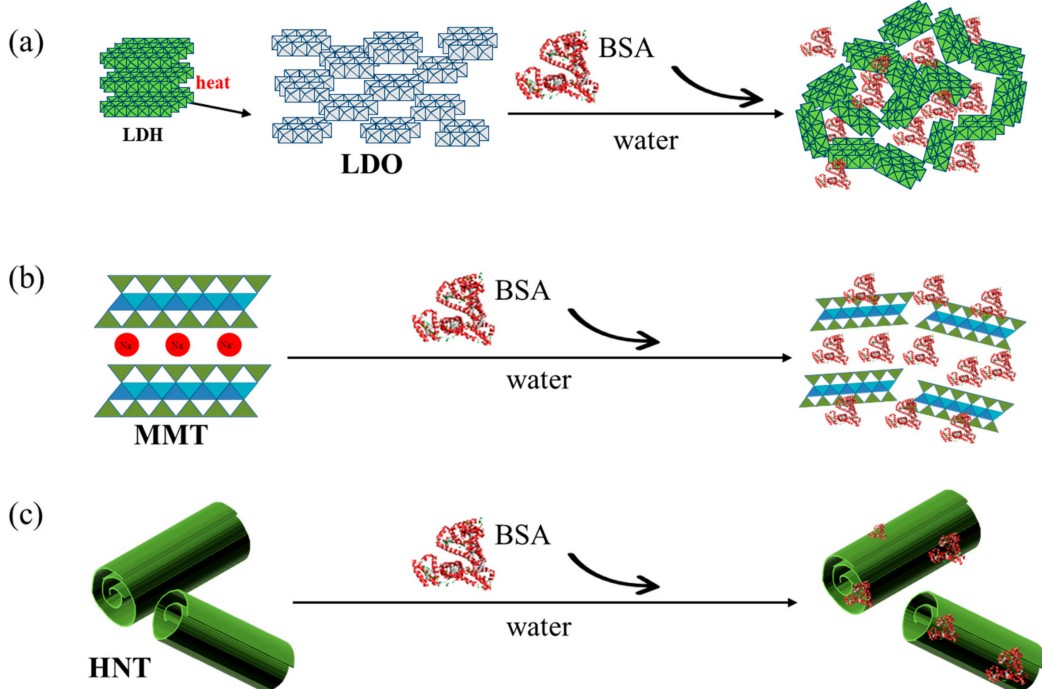

**Figure 1.** Schematic diagram for interactions between bovine serum albumin and clays: (**a**) layered double oxide (LDO), (**b**) Montmorilonite (MMT) and (**c**) Halloysite nanotube (HNT).

As expected, the organic moiety in the protein–clay interactant seemed to occur on the surface of clays. Figure 2 shows the SEM images of clays before and after BSA interactions at different temperatures. It was observed that synthesized LDO had a diameter of about 150 nm, with a coin-like morphology. After BSA interactions, the morphology changed to a sand-rose structure; however, the grain diameter of LDO was not seriously altered, suggesting the proteins did not affect the crystallographic features of this clay. While pristine LDO showed a clear grain boundary in the SEM image, the protein–LDO interactant showed a blurred edge, which was often found when an organic moiety covered the LDO or LDH surface [38,39]. The other two clays showed similar pattern to the LDO–protein interactant in terms of surface morphology. The SEM image of MMT showed random-sized large platelets before the BSA interaction; on the other hand, particle agglomerates with blurred grain boundaries were observed after the BSA interaction. This also suggested delamination of MMT and surface adsorption of the BSA moiety. Pristine particles of HNT were 500 nm in length and less than 100 nm in thickness. After interaction with BSA, we could see that the particles were agglomerated and the boundary of the particles became smooth, suggesting the adsorption of protein moieties on the outer surface of HNT. There was no significant change in the SEM images at 60 °C, implying that the temperature did not affect the nature of the interaction between the clay and protein.

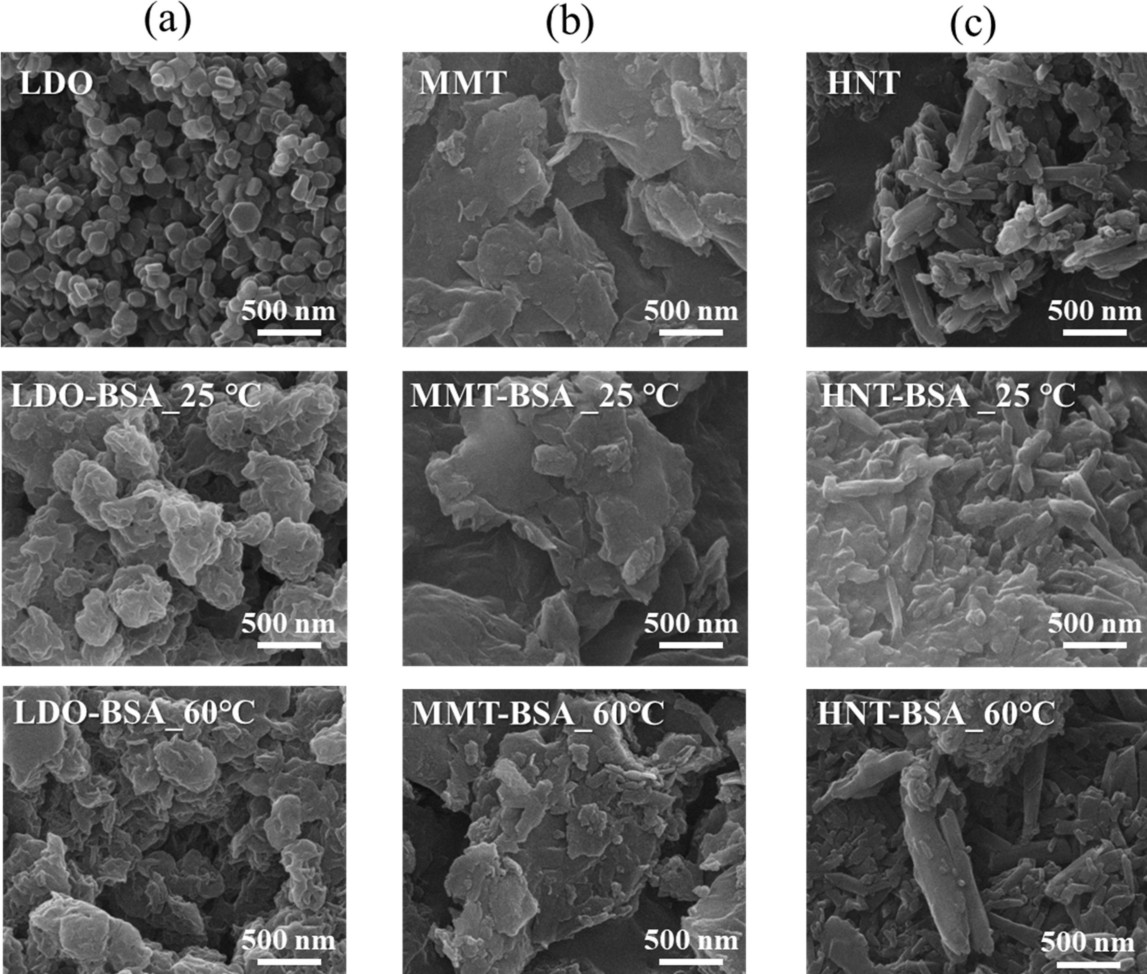

**Figure 2.** Scanning electron microscope image of (**a**) LDO, (**b**) MMT and (**c**) HNT with bovine serum albumin at 25 °C and 60 °C.

In order to confirm the crystallographic change of the clay upon protein interaction, we checked power X-ray diffraction. Figure 3 shows the XRD patterns of the pristine clays before and after BSA interaction at 25 °C and 60 °C. First, LDO showed a typical crystalline phase of periclase (JCPDS No. 45-9846), which is a usual pattern for calcined LDH. Upon interaction with protein in an aqueous condition, the phase transformed to LDH with a carbonate interlayer anion (hydrotalcite; JCPDS No. 22-0700). This result strongly suggested that the BSA molecules did not exist in the interlayer space of LDH type clay; rather the protein moiety exists on the surface or in the inter-particle space. MMT clay showed a typical diffraction pattern for smectite clay (bentonite; JCPDS No. 13-0135) with a slight (001) peak appearing between 2θ 9°~10°. The small (001) peak compared with clear lattice peaks is attributed to the disordered stacking of MMT layers. It is noteworthy that the (001) peak disappeared after the BSA interaction, which suggests the possibility of delamination. As expected from Figure 1, the BSA moiety was in contact with the MMT layer as much as possible, while the crystallographic features of MMT were maintained. The XRD pattern of HNT showed standard halloysite (JCPDS No. 09-0451) before and after the BSA interaction. This strongly suggests that the crystal structure of HNT was not affected by protein and only the surface interacted with protein. As suggested by the previous report [29], it is simple to distinguish whether BSA is attached on the inside lumen or the outside surface. However, we could conclude at least that the BSA did not exist between the lamellar structure of the clays.

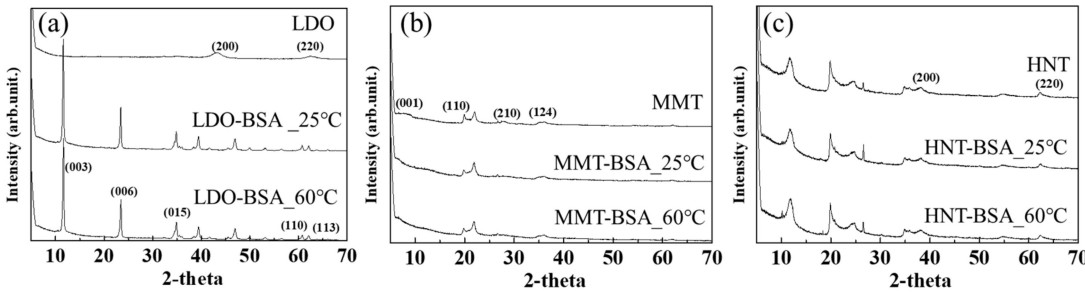

**Figure 3.** X-ray diffraction patterns of (**a**) LDO, (**b**) MMT and (**c**) HNT with bovine serum albumin at 25 °C and 60 °C.

The crystalline phase of HNT and MMT is not altered by the degree of hydration. According to Walid and Marwa [40], MMT showed only lattice expansion or shrinkage (between 13 and 17 Å) along the crystallographic *c*-axis depending on the degree of hydration, while preserving the overall crystalline phase. The effect of water on the crystallinity or morphology of LDO is dramatic. It is well documented in various studies that LDH (hydrotalcite phase) transforms to LDO (periclase phase) through dehydration, dehydroxylation and decomposition of the interlayer anion. The periclase (MgO) is thought to exist in a small domain that is linked by the Al moiety. They are quite vulnerable to hydration and thus the existence of water readily changes the structure. According to Mokhtal et al. [41], the addition of pure water to LDO resulted in meixnerite ($Mg_6Al_2(OH)_{18} \cdot 4(H_2O)$ chemical formula); on the other hand, the addition of water and appropriate anion together gave rise to the original LDH structure. Therefore, the appearance of the LDH (hydrotalcite) phase was expected under the BSA treatment condition in the current research. There are three possible assumptions of the role of BSA: (1) BSA is possibly intercalated in the LDH's interlayer space, (2) atmospheric carbon dioxide dissolved in water resulted in carbonate intercalated LDH with BSA on its surface, or (3) BSA readily adsorbs on the surface of LDO to hinder phase transformation to LDH. Our result showed that the second possibility occurred; BSA did not affect the natural recovery of the LDH structure and it may have only attached on the surface of LDH.

## *3.2. Protein Fluorescence Quenching*

As we confirmed that the proteins would interact at the surface of each clay and that there was no significant change in crystal structure, we evaluated the degree of interaction between BSA and clays utilizing the protein fluorescence quenching assay. This method is fairly powerful for examining the general interaction between exogenous particles and proteins, as the fluorescence is sensitive to a variety of protein conformational changes, such as complexation, aggregation, untangling of three-dimensional structures, etc. [33]. Although the fluorescence quenching assay cannot clearly distinguish the site of denaturation, we can at least quantify the overall interaction between clays and proteins by this means. In terms of surface chemistry, all three clays are fairly hydrophilic due to the surface exposed metal hydroxide (for LDO) and siloxane (for MMT and HNT) groups. Due to the chemical composition, the surface charge is quite different. LDH transformed from LDO usually has a highly positive charge, while both MMT and HNT possess strong negative surface charges over a wide range of pH (Figure 4a). The three clays have different surface adsorbed ions. MMT and LDH have cations and anions, respectively, due to the layer charge; however, HNT does not have typical ions as the halloysite has a neutrally charged layer. The zeta potential values of MMT and HNT moved towards zero (in a positive direction) under the presence of BSA, while that of LDO did not significantly change. These results suggest that MMT and HNT interact with BSA through electrostatic interactions, utilizing the negative surface charge of clays (MMT, HNT) and the positive group of BSA (arginine, glutamine, lysine, histidine, etc.). The interaction between LDO and BSA might also be an electrostatic interaction through the negative group of BSA (glutamate, aspartate, etc.).

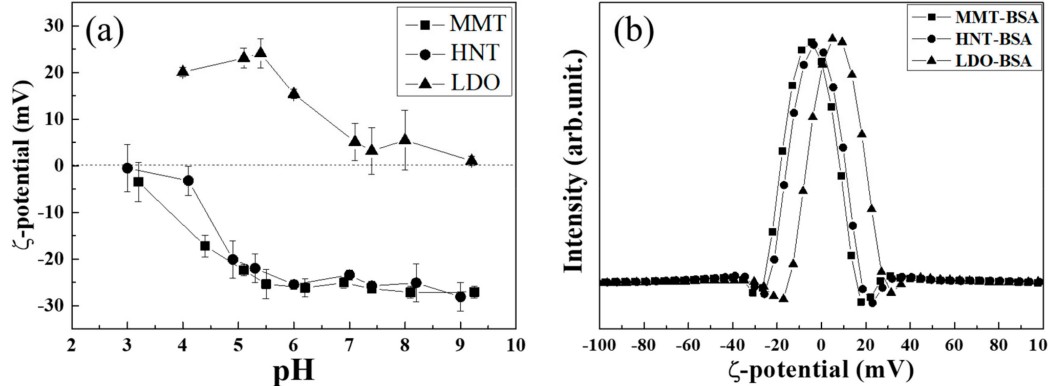

**Figure 4.** (**a**) pH dependent zeta potential of clays and (**b**) zeta potential value distribution pattern of clay with BSA in DPBS.

The fluorescence quenching assay was carried out with varying clay concentrations (Figure 5). All three clays showed less than 30% quenching ratio even at the high clay concentration of 2 mg/mL. This maximum quenching ratio implied that all three tested clays did not interact fatally with BSA as the quenching ratio of other nanoparticles were much larger. Human serum albumin was reported to have a quenching ratio of 69%, 72% and 100% with 2 mg/mL of graphene oxide layer [42], 1.4 mg/mL of 60 nm gold nanoparticles [33] and 0.5 mg/mL of carbon nanotubes [43], respectively. Nevertheless, we could find a difference between MMT and the others in terms of quenching ratio; the maximum quenching ratio at 2.0 mg/mL clay concentration was 27.36, 17.81 and 14.97% for MMT, LDO and HNT.

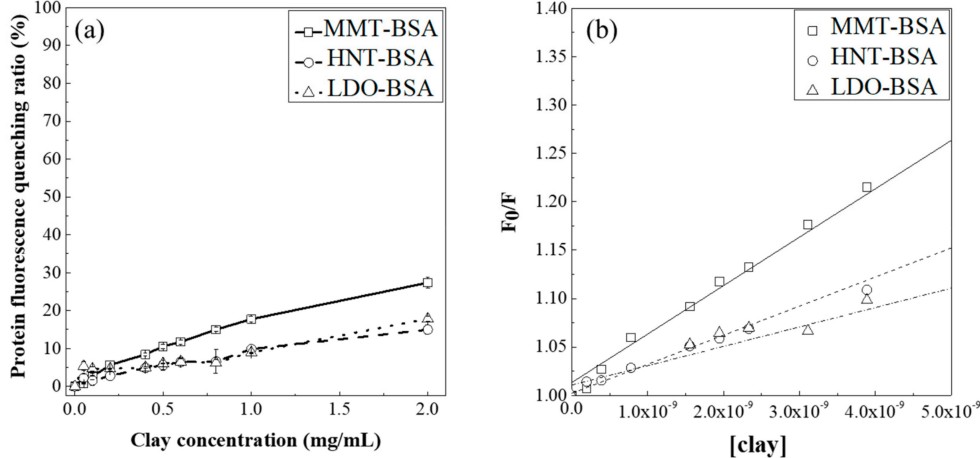

**Figure 5.** (**a**) The protein fluorescence quenching ratio depending on the clay's concentration, (**b**) fitting of the quenching data to the Stern–Volmer equation.

In order to quantitatively investigate the interaction between protein and clay, we fitted the quenching curve to the Stern–Volmer model [33,43], which follows the equation $I_0/I = 1 + k_{sv}[\text{clay}]$ ($I_0$: fluorescence intensity without quencher, $I$: fluorescence intensity with quencher, $k_{sv}$: Stern–Volmer constant, [clay]: concentration of clay suspension). All three curves were well fitted to the equation showing regression factor ($R^2$) values of 0.9899, 0.9847 and 0.9976 for MMT, LDO and HNT treated BSA solutions, respectively. The Stern–Volmer constant, which indicates the binding or association constant between protein–particle interactions, was $4.87 \times 10^7$, $2.68 \times 10^7$ and $2.22 \times 10^7$ [clay]$^{-1}$ for MMT, LDO and HNT, respectively. The degree of interaction between the clays and protein was in the following order: MMT > LDO > HNT. This quantitatively showed that MMT has the strongest effect on the three-dimensional structure of protein, even though the magnitude is not very large. It should be noted here that the fluorescence quenching is related to the adsorption efficiency; the more protein is adsorbed on clay, the higher the chance of an interaction. We evaluated the adsorption amount [44,45]

of BSA on each clay, which was ~150, 80 and 10 mg-BSA/g-clay for MMT, LDO and HNT, respectively, at the same BSA concentration range. Considering both the $k_{sv}$ and adsorption efficiency, HNT was determined to have the least interaction with BSA. However, at this point, we cannot conclude which clay denatures BSA more in terms of its three-dimensional structure.

In addition, we investigated the time and temperature dependent fluorescence quenching behavior under a fixed concentration of clay and protein. A temperature of 60 °C was chosen as it was known that thermal aggregation begins at around this temperature [46,47], and thus we might see clay–protein interactions in different aspects through a temperature dependent study. As shown in Figure 6a, the protein itself did not show any fluorescence quenching at 25 °C. Under the presence of clays, the quenching ratio increased up to 30% and the order was the same as the magnitude of the values. At the elevated temperature of 60 °C, the aspect dramatically changed (Figure 6b). Protein without clay showed a quenching ratio up to 50% due to thermal aggregation. Interestingly, the quenching ratios in the presence of HNT (54%) and LDO (57%) were not significantly higher than that of protein alone, but the value with MMT (80%) was higher, suggesting denaturation of protein was the greatest with MMT under high temperature. Through the Stern–Volmer calculation and temperature experiment, we could confirm that MMT had the strongest clay–protein interaction and HNT had the lowest. LDO took the middle position in terms of adsorption capacity and preservation of protein three-dimensional conformation.

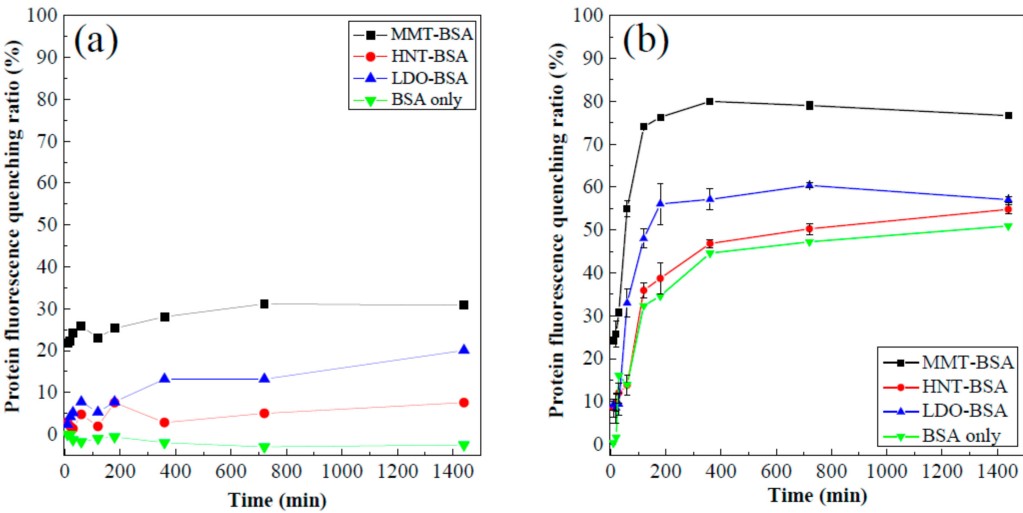

**Figure 6.** Protein fluorescence quenching ratio depending on time at (**a**) 25 °C and (**b**) 60 °C.

As the fluorescence quenching gives us information of the overall denaturation of proteins with clays, we then studied any microscopic change—helix or sheet—utilizing circular dichroism (CD) spectroscopy. It is known that CD spectroscopy is a powerful tool to evaluate the interaction (conformational change or adsorption) between proteins and external materials [30,44,45]. The CD spectrum of albumin alone (Figure 7) showed the positive band at 198 nm originated from the π→π* of a β-sheet and double negative bands at 208 nm and 222 nm, respectively, were attributed to π→π* and n→π* of a pure α-helical structure [31]. The ellipticity value in the spectrum was almost preserved in the presence of clays at 25 °C, except that MMT showed change in the double negative bands. This result is parallel to the recent report that the native-like secondary structure of insulin was well-preserved after incorporation into HNT [27]. The pattern was quite different at an elevated temperature (60 °C); the whole spectrum showed collapse of both negative and positive bands and the degree of collapse was in the following order: MMT-BSA > LDO-BSA > HNT-BSA > BSA alone. The CD result corresponded fairly well to the fluorescence quenching result. The overall denaturation and secondary structure was not seriously affected in the presence of the clays, except MMT. However,

at an elevated temperature, along with thermal aggregation of proteins, clays might influence the three-dimensional structure, as well as the secondary structure, of proteins.

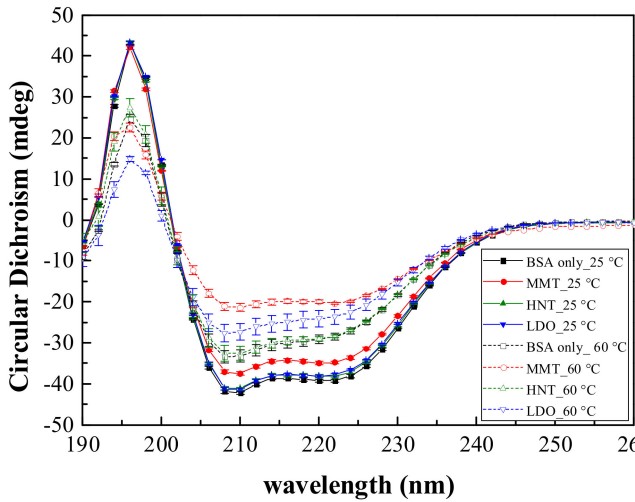

**Figure 7.** CD spectra of BSA with MMT, HNT and LDO.

## 4. Conclusions

Three different kinds of clays; LDO, MMT and HNT, were exposed to albumin and the physico–chemical effect on the protein's conformation was investigated. Protein was expected to be adsorbed on the surface of the clays without altering their structure, and the interaction would be electrostatic between the charged groups of the proteins and the surface charge of the clays. According to the protein fluorescence quenching assay and CD spectroscopy, the three clays did not cause serious denaturation of proteins. Among the three tested clays, MMT had the most influence, possibly due to the high adsorption of BSA on MMT compared with the others. At the elevated temperature of 60 °C, where thermal aggregation of protein begins, the three clays showed some change in fluorescence quenching and CD spectra, and MMT showed the most change among them. Although it is not clear which property of clay is the determining factor on the protein's conformation, it can be concluded that a higher degree of change can be observed with more active surface interactions between clays and proteins, such as with high surface adsorption.

**Author Contributions:** J.-M.O., conceptualization, project administration, writing—original draft preparation; H.-M.K., formal analysis, investigation, writing—review and editing.

**Acknowledgments:** This work was supported by the National Research Foundation of Korea (NRF) grant funded by the Korea government (MSIT) (NO. 2017R1A2B4006352) and (NRF-2017M2A2A6A05093711).

**Conflicts of Interest:** The authors declare no conflict of interest.

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
