# Peer review of "Physico–Chemical Interaction between Clay Minerals and Albumin Protein according to the Type of Clay"

_minerals, doi:10.3390/min9070396_

Reviewer 1 Report

The authors report the physicochemical interaction behavior between three different kinds of clays such as LDO, MMT and HNT and albumin. They found that the protein adsorption amount of clay was different from each other due to the surface charge and swelling property, which is quite interest result and will be useful information for the mineral scientist in the fields of biomedical engineering.

The reviewer recommends that the authors include more references related with biomolecules and clay materials (ex. Journal of Physics and Chemistry of Solids, 67, 1028, 2006; Scientific Reports, 4, 4879, 2014).

Author Response

The authors report the physicochemical interaction behavior between three different kinds of clays such as LDO, MMT and HNT and albumin. They found that the protein adsorption amount of clay was different from each other due to the surface charge and swelling property, which is quite interest result and will be useful information for the mineral scientist in the fields of biomedical engineering. The reviewer recommends that the authors include more references related with biomolecules and clay materials (ex. Journal of Physics and Chemistry of Solids, 67, 1028, 2006; Scientific Reports, 4, 4879, 2014).

=> We appreciate the reviewer’s comment. As the reviewer recommended, we included the suggested references in the introduction part in order to summarize relationship study between biomolecules and clays. (Page 1 line 42)

Reviewer 2 Report

In the present manuscript the authors describe the interaction of the albumin with three different clay minerals, namely montmorillonite (MTT), layered double oxide (LDO) and halloysite nanotubes (HNT), in order to evaluate any denaturation or conformational changes in the protein. They perform their study by using fluorescence quenching assay and circular dichroism.

Although the manuscript could have some potentiality, it lacks of novelty. There are some interesting papers in literature with deal with similar topics and the authors should compare their results with the ones already published (see for example RSC Adv. 2016, 6, 72386-72398, Nanotechnology 2017, 28, 055706, Journal of Colloid and Interface Science 524 (2018) 156–164, Biomacromolecules, 2016, 17, 615-621).

Furthermore the authors could perform additional experiments to support their hypothesis which could add value to their conclusions. I think that the manuscript cannot be accepted for publication in Minerals in its present form and I suggest the authors to re-organize the paper before the publication.

1. The authors report a z-potential study to evaluate the surface charge of the different clays as a function pf te pH. These data are widely reported in literature, thus Figure 4 does not add any claims to the overall manuscript. I suggest the authors to perform the same study in the presence of the albumin, to highlight the interaction of the protein with the clays surface. This could confirm or exclude the interaction site of the albumin in the clay.

2. I think that the fluorescence quenching assay shows that exists an interaction between the protein and the different clays, which is quantified by the value of the Ksv. Before to assert that the protein undergoes denaturation, I think that the authors should perform additional experiments, such as differential scanning calorimetry.

3. How they have quantified the amount of albumin loaded on the clay?

4. What are the measurements units of the Ksv value?

5. The authors should paid more attention in the manuscript preparation, indeed, are present many typos. See for example abstract, lines 39, 57, 62, 148, 198 and so on. Generally it seems that the authors have pasted some part of the manuscript from other sources. Please double check the manuscript and make changes accordingly.

Author Response

Reviewer 2

In the present manuscript the authors describe the interaction of the albumin with three different clay minerals, namely montmorillonite (MTT), layered double oxide (LDO) and halloysite nanotubes (HNT), in order to evaluate any denaturation or conformational changes in the protein. They perform their study by using fluorescence quenching assay and circular dichroism.

Although the manuscript could have some potentiality, it lacks of novelty. There are some interesting papers in literature with deal with similar topics and the authors should compare their results with the ones already published (see for example RSC Adv. 2016, 6, 72386-72398, Nanotechnology 2017, 28, 055706, Journal of Colloid and Interface Science 524 (2018) 156–164, Biomacromolecules, 2016, 17, 615-621).

=> We appreciate the reviewer’s kind suggestions on the references. We read those references on the protein adsorption (immobilization) in HNT ‘s lumen and studied thermal stabilization and conformational change of protein through adsorption on HNT.

We revised both “introduction (page 2 line 47, page 2 line 51)” and “result & discussion (page5 line 181)” sections reflecting the newly added references.

Furthermore the authors could perform additional experiments to support their hypothesis which could add value to their conclusions. I think that the manuscript cannot be accepted for publication in Minerals in its present form and I suggest the authors to re-organize the paper before the publication.

=> We appreciate the reviewer’s kind comment. According to the reviewer’s indication below, we carried out additional experiments to support our hypothesis. The details in experiments and interpretations are as follows.

1. The authors report a z-potential study to evaluate the surface charge of the different clays as a function pf te pH. These data are widely reported in literature, thus Figure 4 does not add any claims to the overall manuscript. I suggest the authors to perform the same study in the presence of the albumin, to highlight the interaction of the protein with the clays surface. This could confirm or exclude the interaction site of the albumin in the clay.

=> We agree to the reviewer’s suggestions on the zeta potential measurement. It is correct that the pH dependent zeta potential is widely known. However, we would like to represent the pH dependent zeta potential value of three clays in order to clarify the surface charge of samples. Furthermore, the zeta potential pattern could be different depending on the origin of clays (HNT, MMT), degree of purification (HNT, MMT), synthesis method (LDO) and etc. Thus it would be helpful for readers to check the real zeta potential values of current samples. In addition, we appreciate the suggestion of the reviewer to examine zeta potential of BSA-clay reactant. We carried out the experiment and added the result as Fig. 4(b) (revised Fig. 4 as below). To summarize, zeta-potential of MMT and HNT clays moved towards zero value (to positive direction) under the existence of BSA, while that of LDO did not significantly changed. The results suggested that the MMT and HNT interact with BSA through electrostatic interaction utilizing negative surface charge of clays (MMT, HNT) and positive group of BSA (arginine, glutamine, lysine, histidine and etc.). The interaction between LDO and BSA might also be electrostatic interaction through negative group of BSA (glutamate, aspartate and etc.).

2. I think that the fluorescence quenching assay shows that exists an interaction between the protein and the different clays, which is quantified by the value of the Ksv. Before to assert that the protein undergoes denaturation, I think that the authors should perform additional experiments, such as differential scanning calorimetry.

=> We appreciate the reviewer’s comments. We carried out DSC measurement (in the range of 30-150°C) for the BSA with or without clays. The calorigram showed exothermal peaks at 96°C, 81°C, 76°C and 68°C for BSA, HNT-BSA, LDO-BSA and MMT-BSA, respectively. It meant that the thermal stability was in the order of MMT-BSA < LDO-BSA < HNT-BSA < BSA alone. The tendency is same to the order obtained from fluorescence quenching; however, it is not clear to discuss interaction between clays and proteins with DSC result. One possible explanation is that the more proteins are adsorbed on clay surface, the more they lose thermal stability. Unfortunately, it is not clear to support clay-protein interaction and we discussed the interaction in terms of adsorption amount instead of representing DSC data.

Fig. S1. Differential scanning calorimetry result for BSA with and without clays, HNT, MMT and LDO

3. How they have quantified the amount of albumin loaded on the clay?

=> We appreciate the reviewer’s comments. We quantified the loaded albumin on clay by measuring initial and final concentration of albumin in the reaction supernatant with Bradford assay. For the application of Bradford assay, the supernatant containing BSA before and after clay reaction were collected by filtering (syringe filter; cellulose membrane). Then 5 mL of supernatant was mixed with 0.1 mL of Bradford dye solution and vortexed. The mixture was incubated at room temperature for 5 mins and absorbance at 595 nm was measured using microplate reader. We represented the procedure in experimental section (page 3 line 109) and clarify the quantification method in the result and discussion part (page7 line 236).

4. What are the measurements units of the Ksv value?

=> As the Stern-Volmer equation is I0/I=1+ksv[clay], the constant has unit of [clay]-1. As the Stern-Volmer constant is usually displayed without unit clarification in many references, we did not indicate the unit. However, we agree to the reviewer’s point that the unit is important, and therefore we clarified the unit of ksv in the manuscript. (Page 7 line 202)

5. The authors should paid more attention in the manuscript preparation, indeed, are present many typos. See for example abstract, lines 39, 57, 62, 148, 198 and so on. Generally it seems that the authors have pasted some part of the manuscript from other sources. Please double check the manuscript and make changes accordingly.

=> We appreciate the reviewer’s consideration. We double-checked the manuscript in order to remove all the typos. In fact, we did not copy and pasted sentences from other literature. However, we thoroughly read the manuscript for finalization.

Reviewer 3 Report

The relevance of the study is clearly presented in the Introduction. Materials and Methods are clearly described.

However, the Results section must be substantially re-written before the manuscript can be published. The following points must be answered:

1. Authors should clearly indicate the pH values, at which circular dichroism (CD) and fluorescence measurements were performed.

2. Authors should explain the reason for conducting CD experiments at  60 deg. C temperature. At such a high temperature, the protein is denatured and forms aggregates with complicated size distribution, and monitoring the interactions of these aggregates with clay is difficult. Authors should explain the purpose of these experiments. How does the distribution of protein aggregates change in this case? The data on the reproducibility of the results obtained in different series of experiments should be presented. Moreover, if experiments at 60 deg. C are discussed, then SEM visualization and X-ray measurements must also be performed for the samples exposed to 60 deg. C temperature.

3. There is a discrepancy in the data presented by the authors, which consists in the following. On the one hand, the authors show that the protein can interact with LDO and HNT clays, with changes in its structure. On the other hand, for some reason there are no changes in the CD spectra, and this discrepancy must be explained. Additionally, error bars should be shown on CD spectra curves.

4. Authors are encouraged to perform spellcheck, and the manuscript must be additionally checked for misprints, which must be corrected ‑ for instance, in Abstract, L. 9 (“minerals”, not “minarals”) and L. 10 (“gastrointestinal medicines”, not “a gastrointestinal medicines”) and in Introduction, L. 51 (“…tertiary structure, which determines the function…”, not “…determine the function…”).

Author Response

1. Authors should clearly indicate the pH values, at which circular dichroism (CD) and fluorescence measurements were performed.

=> We appreciate the reviewer’s kind indication. The pH for CD and fluorescence measurement was set 7.4. We indicated the pH condition for each measurement in the experimental section (Page 3 line 103, 118).

2. Authors should explain the reason for conducting CD experiments at 60 deg. C temperature. At such a high temperature, the protein is denatured and forms aggregates with complicated size distribution, and monitoring the interactions of these aggregates with clay is difficult. Authors should explain the purpose of these experiments. How does the distribution of protein aggregates change in this case? The data on the reproducibility of the results obtained in different series of experiments should be presented. Moreover, if experiments at 60 deg. C are discussed, then SEM visualization and X-ray measurements must also be performed for the samples exposed to 60 deg. C temperature.

=> We fully agree to the reviewer’s point that the specific temperature condition of 60°C should be rationalized. We are going to explain the rationale first in this answer sheet and describe it in the result and discussion session in detail.

1) Experimental condition

The purpose of this experiments is to investigate the effect of clays’ surface on the denaturation of albumin. From previous literature (Journal of pharmaceutical sciences, 2000, 89, 646-651, Plos One, 2016, DOI:10.1371/journal.pone.0153495), thermal aggregation begins at around 60°C. we thought that the temperature point is interesting in terms that the interaction between clays influences aggregation of BSA. As the reviewer suggested, the aggregation resulted in complicated size distribution which hinders quantitative analysis. However, we wish the reviewer and readers understand that this manuscript is to study the effect of clays on protein structure, either denaturation or aggregation, depending on the type of clay. Fluorescence quenching occurs when protein has conformational change including denaturation and aggregation. In this point of view, we could at least claim that the degree of interaction between BSA and clay was in the order of MMT > LDO > HNT.

2) Additional experiments

We fully agree to the reviewer’s point that additional experiments such as X-ray diffraction and electron microscopy measurement at 60°C is helpful for data interpretation. We carried out those experiment and revised Fig. 2 (SEM, images at 60°C was added) and Fig. 3 (XRD, patterns at 60°C was added). To summarize, we could not find significant difference of BSA-clay interactant between room temperature and 60°C treatment in terms of morphology and crystal structure. Although we could not find temperature effect in structure and morphology, we proceeded experiments to investigate 3-dimensional structure of protein structure utilizing fluorescence quenching and CD. Then we found the potential interaction between BSA and clays with the degree of interaction as follows; MMT > LDO > HNT.

3. There is a discrepancy in the data presented by the authors, which consists in the following. On the one hand, the authors show that the protein can interact with LDO and HNT clays, with changes in its structure. On the other hand, for some reason there are no changes in the CD spectra, and this discrepancy must be explained. Additionally, error bars should be shown on CD spectra curves.

=> We appreciate the reviewer’s detailed comment. We agree that our discussion was not clear and there was discrepancy between interpretation. We revised the result and discussion part thoroughly to exclude discrepancy in data interpretation. Throughout the manuscript, we clarified the terminology “interaction” and “denaturation”. Interaction can sometimes give rise to denaturation but cannot affect on protein structure. Furthermore, we clarified that “fluorescence quenching was utilized to find overall interaction between clay and protein”, including a variety of feature (complexation, aggregation, untangling and etc.) and that “CD was utilized more microscopic feature in terms of secondary structure”. To summarize, we found that the more interaction between clay and protein can affect the more on the three dimensional structure and degree of fluorescence quenching was fairly similar to change in CD spectra.

4. Authors are encouraged to perform spellcheck, and the manuscript must be additionally checked for misprints, which must be corrected ‑ for instance, in Abstract, L. 9 (“minerals”, not “minarals”) and L. 10 (“gastrointestinal medicines”, not “a gastrointestinal medicines”) and in Introduction, L. 51 (“…tertiary structure, which determines the function…”, not “…determine the function…”).

=> We appreciate the reviewer’s consideration. We double-checked the manuscript in order to remove all the typos. (Page 1 line 9,10, 56)

Round  2

Reviewer 2 Report

The authors have made some efforts to improve the manuscript quality. I think that the present form is acceptable for publication in Minerals after minor revision.

I suggest the authors to check for example the references list. Some papers are cited two times.

Author Response

=> We appreciate the reviewer's comments. We checked the reference as recommended by the reviewer and revised it. (page 8, line 243)

Reviewer 3 Report

The authors have taken into account my previous comments. However, the manuscript still requires careful revision, as some points still must be clarified. Authors are encouraged to answer the comments listed below.

In my opinion, the mistake made by the authors consists in that they compare initial clay powder with that obtained by drying the clay suspension after its incubation in BSA solutions. Instead of doing this, the authors should compare dried suspensions obtained after incubation of clays in water with those obtained after incubation of clays in BSA solution.

As the authors have noted in the Introduction, clays represent crystalline hydrates, which contain structural water; at that, the amount of structural water is not strictly defined: it depends on the conditions of clay preparation, as well as on the drying time and drying temperature.

This is most clearly seen with the example of LDO. This clay has been prepared by calcinations of LDH at 400 С during 8 hours. (The powdery LDH was calcined at 400 ℃ for 8 h to obtain LDO.) There is no doubt that, upon exposition to pure water, the initial clay powder will capture structural water, what will lead to a change in the crystal structure, as well as in the morphology of the clay.

In Fig. 3а (Figure 3. X-ray diffraction patterns of (a) LDO) it is clearly demonstrated by the authors. The initial powder has two weak peaks ‑ (200) and (220), which disappear after the incubation in BSA solution. At that, however, seven new well pronounced peaks of a crystalline hydrate appear. The interplanar distances corresponding to these peaks are not checked, but the authors attribute their appearance to the influence of BSA.

It is known that BSA molecule denatures at temperatures over 50 С, and, as part of the authors' assumption, one would expect appearance of changes in both the structure and the morphology of the clay after the incubation at 60 С, but this is not observed in the experiments. X-ray diffraction spectra and morphology of the crystals after the incubation of LDO at 20 C and at 60 C are identical. This means that the changes in the structure and the morphology of the clay are caused by incubation of the clay in water rather than by the influence of BSA.

English language still needs improvement, and authors are encouraged to perform the spellcheck carefully.

Author Response

=> We fully agree to the reviewer’s point that degree of hydration is an important parameter to determine crystallinity and morphology of clays, especially for LDH and LDOs. We also found that our standpoint of discussion seemed to under-estimate role of water compared with BSA. Thanks to the reviewer’s considerate review, we could discuss the role of water and BSA at the same time. However, we sincerely ask the reviewer that additional experiments on drying clay suspension (without BSA) instead of initial clay powder may not be necessary as the hydration and clay is well documented in previous works. In the revised manuscript, we added the effect of hydration in crystallinity of clays (HNT, MMT and LDO) and rationalize why we more focused on the action of BSA.

First of all, the crystalline phase of HNT and MMT is not altered depending on the degree of hydration. According to Walid and Marwa (Minerals 2015, 5, 507-526; doi:10.3390/min5030507), MMT showed only lattice expansion or shrinkage (between 13 and 17 Å) along crystallographic c-axis depending on degree of hydration, while preserving overall crystalline phase. Similar to MMT, HNT having kaolinite layer structure showed small change in d-spacing (between 8.6 and 9.4 Å) upon different hydration status with preservation of phase. Thus, we could expect that the XRD patterns of dried suspension of MMT and HNT would be almost same to those of initial powder at appropriate drying condition. In fact, we could not find any significant change in d-spacing of MMT or HNT after BSA treatment in aqueous media, we actually focused on the effect of BSA in crystallinity or morphology.

As clearly pointed out by the reviewer, effect of water in crystallinity or morphology on LDO is dramatic. It is well documented in various literature that LDH (hydrotalcite phase) transformed to LDO (periclase phase) through dehydration, dehydroxylation and decomposition of interlayer anion. The periclase (MgO) is thought to exist in small domain which is linked by Al moiety. They are quite vulnerable to hydration and thus the existence of water readily change the structure. According to Mokhtal, Inayat., et al(Applied Clay Science 50 (2010) 176–181), the treatment of pure water to LDO resulted in meixnerite (‎Mg6Al2(OH)18•4(H2O) Chemical formula); on the other the treatment of water and appropriate anion together gave rise to original LDH structure. Therefore, the appearance of LDH (hydrotalcite) phase was expected under the BSA treatment condition in current research. What we wanted to focus on this paper is the role of BSA in the phase transformation process from LDO to LDH. There are three possible assumptions of the role of BSA: 1) BSA is possibly intercalated in the LDH’s interlayer space, 2) atmospheric carbon dioxide dissolved in water resulted in carbonate intercalated LDH with BSA on their surface, or 3) BSA readily adsorb on the surface of LDO to hinder phase transformation to LDH. Our result showed that case 2) occurred; BSA did not affect the natural recovery of LDH structure and they may be only attached on the surface of LDH.

We sincerely apologize to the reviewer that we omitted important above-mentioned points in discussion of SEM and XRD data. We now realized that readers cannot recognize our point without detailed explanation. Therefore, we revised the manuscript in corresponding result and discussion part.

Round  3

Reviewer 3 Report

The authors have satisfactorily addressed my last comments. Despite my general recommendation to the authors is to perform their research in more detail, now the manuscript can be published.